# PIE-NET: Parametric Inference of Point Cloud Edges

Xiaogang Wang[1,2]    Yuelang Xu[2,5]    Kai Xu[3]    Andrea Tagliasacchi[4]
Bin Zhou[1]    Ali Mahdavi-Amiri[2]    Hao Zhang[2]

[1]State Key Laboratory of Virtual Reality Technology and Systems, Beihang University
[2]Simon Fraser University    [3]National University of Defense Technology
[4]Google Research    [5]Tsinghua University

{wangxiaogang, zhoubin}@buaa.edu.cn; xull16@mails.tsinghua.edu.cn;
{kevin.kai.xu, a.mahdavi.amiri}@gmail.com;
atagliasacchi@google.com; haoz@sfu.ca

## Abstract

We introduce an end-to-end learnable technique to robustly identify feature edges in 3D point cloud data. We represent these edges as a collection of parametric curves (i.e., lines, circles, and B-splines). Accordingly, our deep neural network, coined PIE-NET, is trained for *parametric inference of edges*. The network relies on a region proposal architecture, where a first module proposes an over-complete collection of edge and corner points, and a second module ranks each proposal to decide whether it should be considered. We train and evaluate our method on the ABC dataset, the largest publicly available dataset of CAD models, via ablation studies and compare our results to those produced by traditional (non-learning) processing pipelines, as well as a recent deep learning-based edge detector (EC-Net). Our results significantly improve over the state-of-the-art, both quantitatively and qualitatively, and generalize well to novel shape categories.

## 1 Introduction

Edge estimation is a fundamental problem in image and shape processing. Often regarded as a low-level vision problem, edge detection has been intensely studied and by and large "solved" at the *conceptual* level – there are precise mathematical definitions of what an edge is over an image, or over the surface of a 3D shape. In practice however, even state-of-the-art edge estimators are sensitive to parameter settings and they often underperform near soft edges, noise, and sparse data. This is especially true for acquired point clouds, where these data artifacts are prevalent. We argue this is caused by the fact that edge detection is traditionally achieved by performing decisions based on *manually designed* local surface features; note that this resembles the use of hand-designed descriptors in pre-deep learning computer vision.

In this paper, we advocate for a data-driven approach to feature edge estimation from point clouds – one where priors to make this operation robust are *learned* from training data. More precisely, we develop PIE-NET, a deep neural network that is trained for *Parameter Inference* of feature *Edges* over a 3D point cloud, where the output consists of one or more parametric curves. Our method treats edge inference as a *proposal* and *ranking* problem – a solution that has shown to be extremely effective in computer vision for object detection. More specifically, in a first phase, PIE-NET proposes a large collection of potentially invalid and/or redundant parametric curves, while in a second phase invalid proposals are suppressed, and the final output is generated. The suppression is guided by learnt *confidence* scores estimated by the network, as well as by how well the predicted curves *fit* the data.

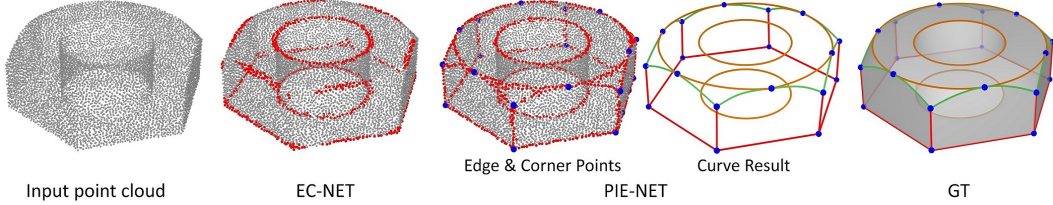

Input point cloud  EC-NET  Edge & Corner Points  PIE-NET  Curve Result  GT

Figure 1: Our deep neural network, PIE-NET, is trained for *parametric inference of edges* from point cloud. It first detects edge and corner points, and then infers a collection of parametric curves representing edge features. In comparison, the only known deep method for this task *only classifies* edge points, and produces results that are inferior to ours both visually and quantitatively.

Our approach is also motivated by recent success on employing neural networks for other low-level geometry processing tasks such as normal estimation [1], denoising [2], and upsampling [3]. We train PIE-NET on the recently released ABC dataset by [4], which is composed of more than one million feature-rich CAD models with parametric edge representations. The combination of large-scale training data and a carefully designed end-to-end learnable pipeline allows PIE-NET to *significantly* outperform traditional (non-learning) edge detection techniques, as well as recent learnable variants from both a quantitative and qualitative standpoint.

## 2   Related work

Literature on point-based graphics [5] is quite extensive and we refer readers to a recent survey [6]. Since the seminal work of Qi et. al. [7], there has been a proliferation of research on learning deep neural networks for 3D point cloud processing [8, 9, 10, 11, 12, 13, 14, 15, 16, 17]. We now focus on methods that are most closely related: ① primitive inference, ② consolidation, and ③ edge detection.

**Parametric primitive inference.** Parametric primitive fitting has been a long-standing problem in geometry processing. The detection or fitting of parametric feature curves (such as Bezier curves) in 3D point clouds has been extensively researched, where it is typically formulated in least squares form [18, 19, 20, 21]. Alternatively, one can use random RANSAC-type algorithms [22] to propose the parameters of primitives fitting a given point cloud. Many types of primitive shapes (planes, spheres, quadratic surfaces) have been considered and various applications have been explored – from 3D reconstruction and modeling [23, 24] to robotic grasping [25]. Besides predefined primitives, recent works also studied learning data-driven geometric priors for shape fitting [26, 27]. End-to-end models have been proposed for fitting cuboids [28], super-quadrics [29], and convexes [30, 31]. Li et al. [11] propose a supervised method to detect a variety of primitive patches at various scales. Similarly to our work, their network first predicts per-point properties, and later estimates primitives.

**Edge-aware consolidation and reconstruction.** Edge feature detection has been applied to enhance 3D reconstruction [32, 33, 34, 35]. Oztireli et al. [32] leverage MLS and robust estimators to automatically identify sharp features and preserve them in the resulting implicit reconstruction. Huang et al. [33] first computes normals reliably away from edges, and then progressively re-samples the point cloud towards edges leading to edge-preserving reconstruction. Edge detection has also been utilized to enhance RGBD reconstruction, where Liu et al. [35] propose to detect edges for the task of wire reconstruction from RGBD sequences. Consolidation has also been adopted in designing deep neural networks. Yu et al. [36] propose PU-Net for point cloud upsampling. It first learns per point features and then expands them with a multi-branch convolution unit. The expanded feature is then split to a multitude of features used to reconstruct a dense point set. EC-Net [10] proposes a deep edge-aware point cloud consolidation framework. The network is trained to regress and recover upsampled 3D points and point-to-edge distances. The reconstruction of surfaces with sharp edges has also recently been tackled by learnable sparse convex decomposition [31], but this method performs a feature-aware reconstruction as a *holistic* task.

**Edge feature detection.** Edge feature detection from point clouds relies on local geometric properties such as normals [37], curvatures [38], and feature anisotropy [39]. For example, Fleishman et al. [40] employ robust estimators to identify edge features, but shape analysis relies on moving least squares (MLS) to locally model the neighborhood of a point. Daniels et al. [41] extend this method to extract feature *curves* from noisy point clouds based on the reconstructed MLS surface. It is also

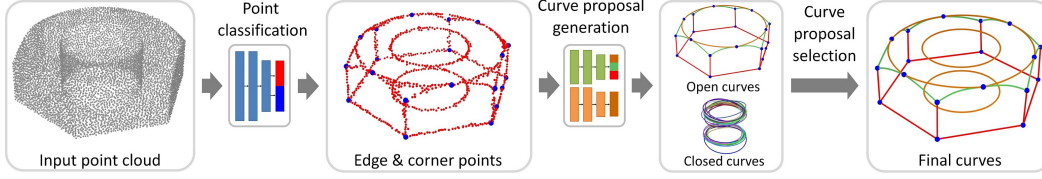

Figure 2: **Pipeline** – Our method treats parametric edge inference as a region proposal task. Given a point cloud, our network first detects edges and corners. Then, for each pair of corners, it performs curve proposal generation and selection to detect feature edges as a set of parametric curves.

possible to extract edge features via point cloud segmentation of these local properties [42, 43]. The closest work to ours is EC-Net [10], whose first phase consists of point classification and regression of per-point distances to the edge. Edge points are then detected as the points with a zero point-to-edge distance. To the best of our knowledge, EC-Net is the only prior work on feature edge estimation using deep learning and PIE-NET is the first technique that estimates parametric edge curves with an end-to-end trainable deep network.

## 3 Method

The processing pipeline of **P**arametric **I**nference of **E**dges **Net**work is summarized in Figure 2. Our technique treats point cloud curve inference as curve *proposal* process followed by *selection*, a technique inspired by image-based object detection pipelines [44].

**Overview.** Given a point cloud, a detection module first identifies *edge* and *corner* points (Section 3.1). Corners represent the start/end points of curves, or the locations where two curves touch. Pairs of corners are then given to a curve proposal module (Section 3.2) to identify the corresponding "edge points", and finally generate a corresponding parametric open curve proposal. For closed curves, as they do not have a well defined start/end, we take inspiration from the Similarity Group Proposal Network [45], and regress the parameters of closed curves by first performing a clustering of the identified edge points, followed by curve fitting. Finally, a simple curve selection scheme merges all the curve proposals to generate the final fitting result.

**Training data.** We performed a statistical analysis over the ABC dataset [46], a large-scale CAD mechanical part dataset containing ground-truth annotations for edges and corner points. We found that the edges of the mechanical parts largely belong to three types, i.e., lines, circles, and B-spline curves, which account for more than 95% of the edges. We ignored the "ellipse" and "other" types due to their statistical insignificance. Therefore, we only focus on these three curve types in this paper, and filter out other types in the ground truth. For each ABC model, we sample it into a point cloud containing 8,096 points, via uniform point sampling. We then transfer the ground-truth annotations of the CAD models to the point clouds by nearest neighbor assignment.

**Parameterization.** Our method deals with three types of curves: lines, circles (open arcs plus full closed circles), and B-splines. We now describe the differentiable parameterization of the two latter curve types, where we note that a line segment can be formed by simply connecting two corner points. We parameterize circles as $\beta=(p_1, p_2, p_3)$, where $p_1, p_2, p_3$ are any three points on the circle that are not collinear. We first transform $\beta=(p_1, p_2, p_3)$ as $(n, c, r)$, where $n$ is the normal of the circle, and $c$ and $r$ are its center and radius. We then randomly sample points on the circle, and express them as a function of $\beta$. To achieve this, we draw $\alpha \in [0, 2\pi]$, and generate random samples $p(\alpha|\beta)=c + r(u\cos(\alpha) + v\sin(\alpha))$, where $u=p_1 - c$ and $v=u \times n$. We parameterize B-Spline curves with four control points $\beta=\{p_i\}_{i=0}^3$. Given $B_{i,K}(\cdot)$ representing the $i$-th basis function of a $K$-th order B-spline, and $\alpha$ sampled uniformly in the $[0, 1]$ range, we draw a uniform random sample as $p(\alpha|\beta)=\Sigma_i p_i B_{i,K}(\alpha)$. Note that to ensure that $p(0|\beta)=c_1$ and $p(1|\beta)=c_2$, we employ *quasi-uniform* B-splines. We predict residuals as the *displacement* of the two intermediate control points $\{p_i\}_{i=1,2}$.

### 3.1 Point classification

To classify points in the input point cloud into edge and corners (plus the null class), we use a PointNet++ like architecture [47]. In particular, we devise two separate classification networks outputting edge vs. null and corner vs. null, respectively. The network predicts the probability of

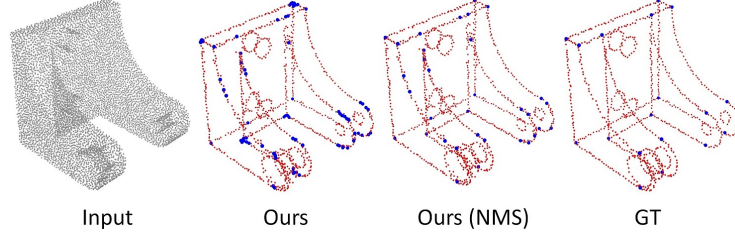

| Input | Ours | Ours (NMS) | GT |

Figure 3: **Point classification** – We determine whether each point belongs to an edge or to a corner. We show the results of our method before and after NMS, as well as the corresponding ground truth.

a point being on an edge $T_e$, or a corner $T_c$, or null otherwise, as well as the 3D *offset vector* $D_e$ that projects the point onto an edge, and the offset $D_c$ that projects the point onto a corner – these predictions are supervised by the ground truth labels $\hat{T}_e$ and $\hat{T}_c$, as well as the ground truth offsets $\hat{D}_e$ and $\hat{D}_c$. We then threshold the probabilities $T_e > \tau_e$ and $T_c > \tau_c$ to obtain a corresponding set of edge points $E = \{e_i\}_{i=1}^{M}$ and corner points $C = \{c_i\}_{i=1}^{N}$. We set the parameters $\tau_e = 0.7$ and $\tau_c = 0.9$ throughout our experiments. We optimize the multi-task loss $\mathcal{L}_{\text{detection}} = \mathcal{L}_{\text{edge}} + \mathcal{L}_{\text{corner}}$:

$$\mathcal{L}_{\text{edge}} = \mathcal{L}_{\text{cls}}(T_e, \hat{T}_e) + \hat{T}_e \cdot \lambda_e \mathcal{L}_{\text{reg}}(D_e, \hat{D}_e), \tag{1}$$

$$\mathcal{L}_{\text{corner}} = \mathcal{L}_{\text{cls}}(T_c, \hat{T}_c) + \hat{T}_c \cdot \lambda_c \mathcal{L}_{\text{reg}}(D_c, \hat{D}_c), \tag{2}$$

where for $\mathcal{L}_{\text{cls}}$, rather than traditional binary cross-entropy, we employ a *focal loss* [48] to deal with the pathological class imbalance in our dataset (the number of edge points is very small relative to the number of non-edge points). For $\mathcal{L}_{reg}$, we use the *smooth $L_1$* loss as in [49, 44].

**Non-maximal suppression.** At test time, we first classify corner points and regress the corresponding offset vectors. However, due to noise, several points may be mislabelled as corners in the proximity of a ground-truth corner, hence necessitating post-processing. To this end, we adopt a point-level Non-Maximal Suppression (NMS). After applying the offset to the detected corner points, we perform agglomerative clustering with a maximum intra-class distance threshold $\delta$; we use $\delta = 0.05\ell$, with $\ell$ being the diagonal length of object bounding box. We then perform NMS within each cluster by selecting the corner with the highest classification probability; see Figure 3.

## 3.2 Open Curve proposal

PIE-NET performs a curve proposal for *open* curves and another for *closed* curves. Our *open* curve proposal generation leverages the fact that open curves connect two corner points, and makes the assumption that a corner pair can support at most one curve. Hence, given the set $C$ with $N$ corner points, we start by generating all $O(N^2)$ corner point combinations and create corner pairs $\{\mathcal{P}_i = \{c_{i1}, c_{i2}\} \mid c_{i1}, c_{i2} \in C\}$. Each corner pair will correspond to a curve proposed by the curve proposal generation; see Figure 4. Given a pair $\mathcal{P}_i$, we need to identify points that should be associated to the corresponding curve. We first localize the search via a heuristic that only considers points in $E$ that lie within a sphere with center $(c_{i1} + c_{i2})/2$, and radius $R = \|c_{i1} - c_{i2}\|/2$. Within this sphere, we then uniformly sample a subset $E_i^o$ that has a cardinality compatible with the input dimension of our multi-headed PointNet networks and feed $E_i^o$ to them; see Figure 4.

**Losses.** The network heads perform three different tasks, and are trained by three different losses. The first network head performs *segmentation*, determining whether a particular point belongs to the candidate curve. The second head performs *classification*, determining whether we need to generate a line, circle, or B-spline. The third head performs *regression*, identifying the parameters of the proposed curve. Note that the network outputs the parameters for all curve types and only those corresponding to the type output by the type classifier are regarded as the valid output. More formally:

$$\mathcal{L}_{\text{proposal}} = w_{\text{m}} \mathcal{L}_{\text{mask}}(M_p, \hat{M}_p) + w_{\text{c}} \mathcal{L}_{\text{cls}}(T_p, \hat{T}_p) + w_{\text{p}} \mathcal{L}_{\text{para}}(\beta), \tag{3}$$

where $M_p$ and $\hat{M}_p$ are the predicted / ground-truth classifications, $T_p$ and $\hat{T}_p$ are predicted / ground-truth curve types, and $\beta$ represents the predicted curve parameters. We set $w_{\text{m}} = 1$, $w_{\text{c}} = 1$, and $w_{\text{p}} = 10$ throughout our experiments. Softmax cross-entropy is employed for both $\mathcal{L}_{\text{mask}}(.)$ and $\mathcal{L}_{\text{cls}}(.)$, while for regression:

$$\mathcal{L}_{\text{para}} = \hat{T}_{\text{circle}} \cdot \mathcal{L}_{\text{circle}}(\beta) + \hat{T}_{\text{line}} \cdot \mathcal{L}_{\text{line}}(\beta) + \hat{T}_{\text{spline}} \cdot \mathcal{L}_{\text{spline}}(\beta), \tag{4}$$

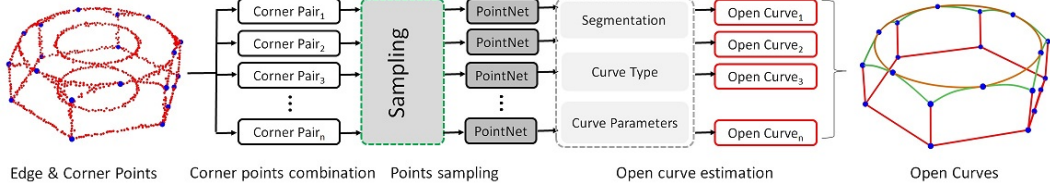

<div align="center">Edge & Corner Points    Corner points combination  Points sampling        Open curve estimation        Open Curves</div>

Figure 4: **Open curve proposal** – Given the collection of corners, we generate all possible corner pairs and propose an open curve connecting the pair of corners. In doing so, we predict the curve type, the parameters, and the subset of points corresponding to the proposed curve.

where $\hat{T}_*$ encodes the ground truth one-hot labels for the corresponding curve type, while $\mathcal{L}_*$ are Chamfer Distance losses measuring the expectation of Euclidean distance deviation between the curve having parameters $\beta$ and the ground truth. In order to compute expectations, we need to draw random samples from each curve type which are *parametric* in the degrees of freedom $\beta$.

### 3.3 Closed curve proposal

Our closed curve proposal module is inspired by [45]. In particular, we first identify the subset of points belonging to closed curves via feature *clustering*, and then *fit* a closed curve to each proposed cluster, while simultaneously estimating the *confidence* of the fit; see Figure 5. Here we use the edge/corner classification described in Section 3.1 of the main paper. Note our method currently only handles curves with a *circular* profile, but it can be easily extended to other types of closed curves.

**Clustering.** We train an equivariant PointNet++ network to produce a point-wise feature $F(\cdot)$ for each of the $M$ input edge points. Based on such features, we then create a similarity matrix $S \in \mathbb{R}^{M \times M}$, where $S_{ij} = \|F(p_i) - F(p_j)\|_2$. We can then interpret each of the $M$ rows of $S$ as a *proposal*, and consider the set $C_m = \{j \text{ s.t. } S_{m,j} < \bar{S}\}$ as the edge points of the $m$-th proposal. $\bar{S}$ is a threshold to filter out the points attaining very different feature scores (i.e., they probably do not belong to the same curve). Potentially redundant proposals are dealt with in the *selection* phase of our pipeline, as described in Section 3.4.

**Fitting.** We take each proposal $C_m$, and regress the parameters $\beta$ of the corresponding curve, as well as its confidence $\gamma$. We parameterize each circle proposal via three points $\beta = \{p_a + \Delta_a, p_b + \Delta_b, p_c + \Delta_c\}$. We obtain $\{p_a, p_b, p_c\}$ by furthest point sampling in $C_m$ initialized with $p_a = p_m$. We then train a PointNet architecture with two fully-connected heads. The first head regresses the offsets $\{\Delta_a, \Delta_b, \Delta_c\}$, while the second head predicts $\gamma$.

**Losses.** We train our network to predict similarity matrices $S$ given ground truth $\hat{S}$, confidences $\Gamma = \{\gamma_n\}$ given ground truth $\hat{\Gamma}$, and a collection of points sampling the ground truth curve:

$$\mathcal{L}_{\text{closed}} = \mathcal{L}_{\text{sim}}(S, \hat{S}) + \mathcal{L}_{\text{score}}(\Gamma, \hat{\Gamma}) + \mathcal{L}_{\text{para}}(\beta). \tag{5}$$

Given that $\hat{S}_{ij} = 0$ if points $p_i$ and $p_j$ belong to the same ground truth curve and $\hat{S}_{ij} = 1$ otherwise, we supervise for similarity via:

$$\mathcal{L}_{\text{sim}} = \sum_{ij} S_{ij}, \tag{6}$$

where

$$S_{ij} = \begin{cases} \|F(p_i) - F(p_j)\|_2, & \hat{S}_{ij} = 1 \\ \max\{0, K - \|F(p_i) - F(p_j)\|_2\}, & \hat{S}_{ij} = 0 \end{cases}$$

where $F$ is point-wise feature computed with PointNet++. $K$ controls the dissimilarity between elements in different parts, which is set to $K = 100$ in our experiments. For $\mathcal{L}_{\text{score}}(\cdot)$ we employ $L_2$ loss, where $\hat{\Gamma}$ is the segmentation confidence. Positive training examples come from seed points belonging to ground truth closed curves, and their IoU with ground truth segmentations is larger than $0.5$. Negative training examples are those with IoU smaller than $0.5$. For parameter regression, we minimize

$$\mathcal{L}_{\text{para}} = \hat{T}_{\text{circle}} \cdot \mathcal{L}_{\text{circle}}(\beta), \tag{7}$$

where $\hat{T}_{\text{circle}}$ is the ground truth one-hot labels, and $\mathcal{L}_{\text{circle}}(\cdot)$ is the Chamfer distance between the curve represented by $\beta$ and the ground truth. In particular, we first compute the circle according to the estimated $\beta$, and then sample it by points. We then compute the Chamfer distance between the estimated circle and its corresponding point-sampled ground-truth.

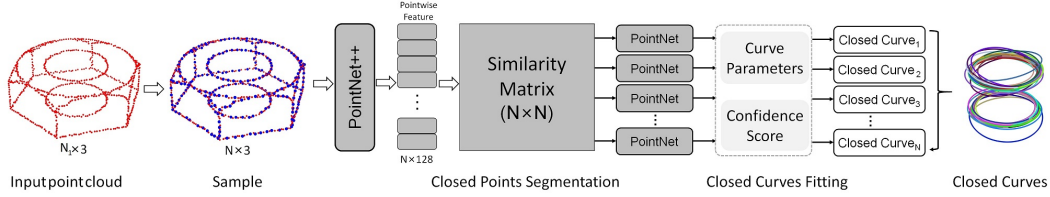

Figure 5: **Closed curve proposal** – We first identify the subset of points belonging to closed curves via feature-based clustering and then fit a closed curve to each cluster. The network outputs both curve parameters and confidence scores.

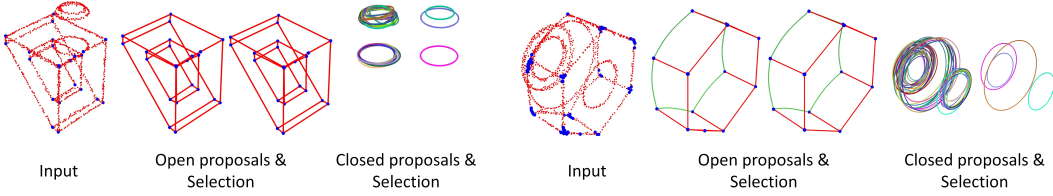

Figure 6: **Proposal selection** – The curves generated by open/closed proposals, and the ones that were accepted by our selection process.

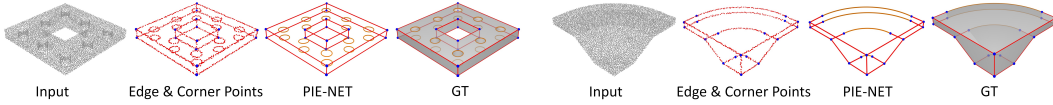

Figure 7: Results on edge and corner detection and parametric curve inference by PIE-NET.

### 3.4 Curve proposal selection

Similar to proposal-based object detection for images [44], the final stage of our algorithm is a non-differentiable process for redundant/invalid proposal filtering; see Figure 6. We adopt slightly different solutions for *open* and *closed* curves.

**Open curve selection.** Given the segmentations, i.e., a set of points associated with a curve (see Figure 5 in the main paper) corresponding with two proposals, we first measure overlap via $O(A, B) = max\{I(A, B)/A, I(A, B)/B\}$, where $I(A, B)$ is the cardinality of the intersection between the sets. We then merge the two candidates, if $O(A, B) > \tau_o$ and retain the curve with larger cardinality, where we use $\tau_o = 0.8$ as determined by hyper-parameter tuning.

**Closed curve selection.** The similarity matrix produces a closed-curve proposal for *each* of its $N$ rows. Even after discarding proposals with confidence score $\gamma_n < \tau_\gamma$, many are *non-closed* curves, or represent the *same* closed curve; see Figure 6. We perform agglomerative clustering for proposals when $\text{IoU}(A, B) > \tau_{\text{iou}}$, and retain the proposal in the cluster with the highest confidence. We use $\tau_\gamma = 0.6$ and $\tau_{\text{iou}} = 0.6$ for all our experiments. Finally, for each closed segment, we select the best matching closed curve. Specifically, we use the Chamfer Distance to measure the matching score.

## 4 Results and evaluation

In Figure 7, we show randomly selected qualitative results produced by PIE-NET, and generalization to object categories that are not part of our training dataset in Figure 10. We evaluate our network via ablation studies (Section 4.1), comparisons to both *traditional* and *learned* pipelines for edge detection (Section 4.2), and stress tests with respect to noise and sampling density (Section 4.3). To evaluate edge classification, we measure precision/recall and the IoU between predictions, while to evaluate the geometric accuracy of the reconstructed edges, we employ the *Edge Chamfer Distance* (ECD) introduced by [31]. Note that, differently from Chen et al. [31], we *do not* need to process the dataset to identify ground-truth edges, as these are provided by the dataset.

| Metric | ① | | ② | | ③ | | ④ | |
| | $R' = 1.5R$ | $R' = 3R$ | $\tau_c = 0.8$ | $\tau_c = 0.95$ | $\tau_e = 0.6$ | $\tau_e = 0.8$ | w/o $D_e, D_c$ | PIE-NET |
|---|---|---|---|---|---|---|---|---|
| ECD ↓ | 0.0326 | 0.0824 | 0.0150 | 0.0144 | 0.0163 | 0.0149 | 0.0186 | **0.0136** |
| IOU ↑ | 0.4386 | 0.2950 | 0.4875 | 0.5110 | 0.4356 | 0.4964 | 0.5017 | **0.5330** |
| Precision ↑ | 0.5149 | 0.3244 | 0.5700 | 0.6032 | 0.5180 | 0.5975 | 0.5824 | **0.6219** |
| Recall ↑ | 0.8570 | **0.8910** | 0.8415 | 0.8230 | 0.8340 | 0.8035 | 0.7849 | 0.8165 |

Figure 8: **Ablation studies** – We evaluate the qualitative (top) and quantitative (bottom) performance of our method across a number of metrics as we tweak: ① the radius $R$ controlling the sampling heuristic, ② the corner segmentation threshold $\tau_c$, ③ the edge segmentation threshold $\tau_e$, and ④ the edge and corner offsets $D_e$ and $D_c$. Recall for PIE-NET: $R'=R$, $\tau_c=.9$, $\tau_e=.7$, and we use $D_e, D_c$.

| | VCM | | | EAR | | | EC-Net | PIE-NET | PIE-NET |
| | $\tau=0.12$ | $\tau=0.17$ | $\tau=0.22$ | $\tau=0.03$ | $\tau=0.035$ | $\tau=0.04$ | | RS | |
|---|---|---|---|---|---|---|---|---|---|
| ECD ↓ | 0.0321 | 0.0430 | 0.0569 | 0.0679 | 0.0696 | 0.0864 | 0.0360 | 0.0137 | **0.0088** |
| IOU ↑ | 0.2841 | 0.2854 | 0.2855 | 0.3404 | 0.3250 | 0.2844 | 0.3561 | 0.5976 | **0.6223** |
| Precision ↑ | 0.3063 | 0.3244 | 0.3456 | 0.5560 | 0.4149 | 0.6523 | 0.4872 | 0.6816 | **0.6918** |
| Recall ↑ | 0.8385 | 0.7644 | 0.6937 | 0.4820 | 0.5910 | 0.3578 | 0.5736 | 0.8319 | **0.8584** |

Figure 9: **Comparisons to state-of-the-art methods** – Qualitative (top) and quantitative (bottom) comparisons against point cloud with random sampling (RS) and edge detection techniques – VCM [50], EAR [51], and EC-Net [10].

## 4.1 Ablation studies

**Sphere radius – $R$.** We validate the spherical sub-sampling heuristic introduced in Section 3.2. Note that we only consider values of $R'$ strictly larger than the default setting, as otherwise we are *guaranteed* to miss edge features. Specifically, we set $R'$ at ×1.5 and ×3 of the default $R$ – in the limit ($R'=\infty$) we would consider the *entire* point cloud for each candidate corner-pair. Our analysis shows that as the sampled point cloud becomes larger and larger, it becomes more and more difficult to identify the right subset of points. This is because our sphere sampling heuristic also provides a *hint* of which curve needs to be sampled – the one whose corner points are touching the sphere.

**Classification thresholds – $\tau_c, \tau_e$.** We also study curve generation performance as we vary how many corner points are accepted. Overall, the performance of the network is stable as we vary $\tau_c$, delivering the expected precision vs. recall trade-off. We find that curve generation quality slightly improves when we increase $\tau_c$, but as the threshold gets too large, the quality begins to decline as several corner points are filtered out, resulting in the absence of some feature curves. Analogous trade-offs are observed as we adjust the values of edge classification threshold $\tau_e$.

## 4.2 Comparisons to the state-of-the-art

Two of the most classical, pre-deep learning, methods for point cloud edge detection are: ① Merigot et al. [50], where edges are detected by thresholding the Voronoi Covariance Measure (VCM), and ② Huang et al. [51], where edges are identified as part of an Edge-Aware Resampling (EAR) routine. We consider these two methods as representative of the state of the art, as both have been adopted in the point-set processing routines of the well known CGAL library [52]. As reported in Figure 9, our end-to-end method completely outperforms these classical baselines.

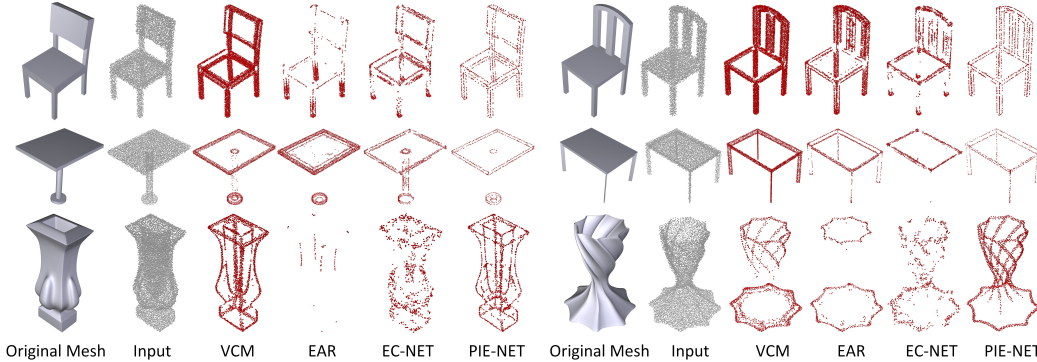

Original Mesh    Input    VCM    EAR    EC-NET    PIE-NET    Original Mesh    Input    VCM    EAR    EC-NET    PIE-NET

Figure 10: **Generalization to novel object categories** – While PIE-NET is trained on CAD models of mechanical assemblies from the ABC dataset [46], our edge detector is immediately applicable to 3D point clouds of general 3D objects and consistently outperforms VCM, EAR, and EC-Net.

**Voronoi Covariance Measure (VCM) [50].** We compute VCM for each point VCM($p_i$), where we set *offset radius* to $0.5$ and *convolution radius* to $0.25$. These parameters were found by a parameter sweep evaluated over the test set of ABC [4]. We then consider a point to belong to an edge if VCM($p_i$)$>\tau$, and select three pareto-optimal thresholds $\tau$=$\{0.12, 0.17, 0.22\}$ for evaluation.

**Edge Aware Resampling (EAR) [51].** This method first re-samples away from edges via anisotropic locally optimal projection (LOP), an operation that leaves *gaps* near sharp edges. We classify points as edges by detecting whether they fall within these gaps. This is achieved by computing the average distance $D_{10}(p_i)$ to the ten nearest neighbors – we consider a point to belong to an edge if $D_{10}(p_i) > \tau$. We report the results for $\tau = \{0.03, 0.035, 0.04\}$ – the best performing thresholds as selected by a dense sweep in the $[0.01 \ldots 0.05]$ range as suggested in the paper. As illustrated in Figure 9, our end-to-end method performs significantly better, as immediately quantified by the fact that ECD is an *order of magnitude* larger than the one reported for PIE-NET.

**Edge-Aware Consolidation Network (EC-Net) [10].** We train EC-Net on our dataset, not the author's dataset [10] as it only contained 24 CAD models with manually annotated edges, while ABC contains more than a million models. The comparison results demonstrate how PIE-NET achieves significantly better performance; see Figure 9. In particular, our qualitative analysis revealed how EC-Net struggles in capturing areas with *weak-curvature*, as well as *short* edges.

In Figure 10, we show additional qualitative comparison results on *novel* object categories such as tables, chairs, and vases, etc., which are *not* part of the ABC training set. These results were obtained using the same methods and trained networks as those that produced Figure 9. The point cloud inputs were obtained by uniform sampling on the original mesh shapes, which are shown in Figure 10 to reveal the edge features; there were no GT edges associated with these shapes.

## 4.3    Stress tests

**Random Sampling (RS).** We sampled 100K points uniformly over each CAD shape, and then sub-sampled $8,096$ points non-uniformly via random sampling. We re-trained and re-tested PIE-NET on the new point clouds, keeping all other settings unchanged. The performance is reported in Figure Figure 9. As we can see, while there is a slight performance degradation, they are quite comparable to original and still outperform all performance statistics obtained by VCM, EAR, and EC-Net, except for one case, VCM with $\tau = 0.12$, which yielded a recall of $0.8385$, but it is paired with a very low precision of only $0.3063$.

**Point cloud noise.** We stress test PIE-NET by increasing the level of noise. Specifically, we randomly apply different perturbations to the point samples along the surface normal direction with a scale factor in the $[1.0 - X, 1.0 + X]$ range, where we tested four values of $X$=$\{0, 0.01, 0.02, 0.05\}$. In each case, the network was trained with the noise-added data. Figure 11 shows some visual results and quantitative measures. As we can observe, our network, even when trained with noisy data, can still out-perform VCM, EAR, and EC-Net when they are tested on or trained on clean data.

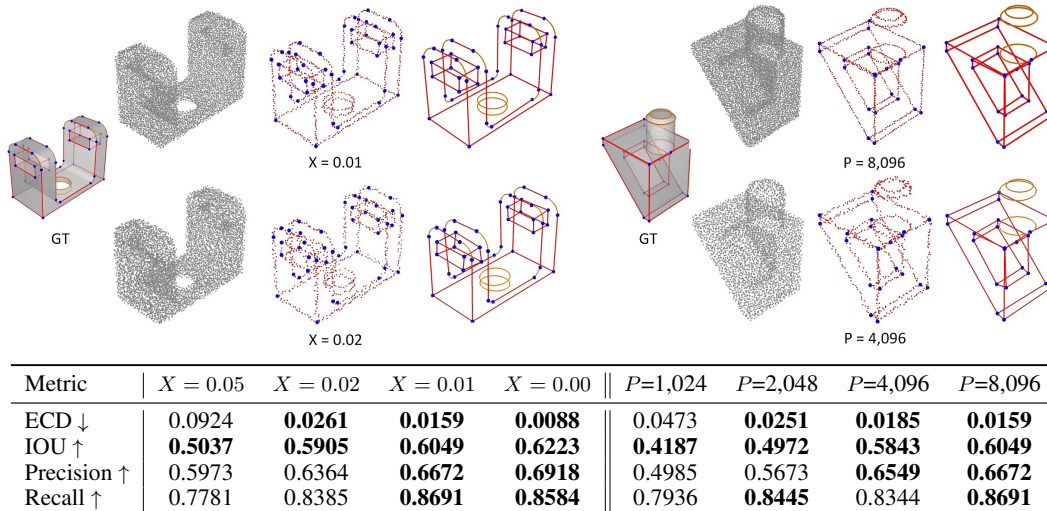

| Metric | $X=0.05$ | $X=0.02$ | $X=0.01$ | $X=0.00$ | $P$=1,024 | $P$=2,048 | $P$=4,096 | $P$=8,096 |
|---|---|---|---|---|---|---|---|---|
| ECD ↓ | 0.0924 | **0.0261** | **0.0159** | **0.0088** | 0.0473 | **0.0251** | **0.0185** | **0.0159** |
| IOU ↑ | **0.5037** | **0.5905** | **0.6049** | **0.6223** | 0.4187 | 0.4972 | 0.5843 | 0.6049 |
| Precision ↑ | 0.5973 | 0.6364 | **0.6672** | **0.6918** | 0.4985 | 0.5673 | **0.6549** | **0.6672** |
| Recall ↑ | 0.7781 | 0.8385 | **0.8691** | **0.8584** | 0.7936 | **0.8445** | 0.8344 | **0.8691** |

Figure 11: **Network behavior with respect to noise and sampling density** – We show the qualitative (top) and quantitative (bottom) performance of PIE-NET on input point clouds degraded by noise of different magnitudes: $X$={0.05, 0.02, 0.01, 0} or sampled at a reduced density: with $P$ going from 8,096 down to 1,024. All the numbers in **boldface** outperform VCM, EAR, and EC-Net without added noise or reduced point density; see the table in Figure 9 for reference.

**Point density.** We also train PIE-NET on point clouds at a reduced density. Specifically, for each CAD shape, we sampled a different number ($P$) of points to verify whether our network could handle the sparser point clouds, where $P$={8,096, 4,096, 2,048, 1,024}. Results in Figure 11 reveal a similar trend as from the previous stress test. Namely, our network, when trained on sparser point clouds, can still outperform VCM, EAR, and EC-Net when they are tested on or trained on data at full resolution (8,096 points).

## 4.4 Timing

For all the results shown in the paper, the average running time is about $0.5$ second for point classification and 3 seconds for curve generation, per point cloud. In comparison, the average running times for point classification by VCM, EAR, and EC-Net are $5.5$, $4.0$, and $0.8$ seconds, respectively – they are all slower than PIE-NET. Training times for point classification, the open and closed curve proposal networks were about 23, 12, and 8 hours, respectively, for 100 epochs, on an NVIDIA TITIAN X GPU.

## 5 Conclusion

The detection of edge features in images has been shown to play a fundamental role in low-level computer vision — for example, the first layers in deep CNNs have been shown to be nothing but edge detectors. Notwithstanding, limited work to detect features of visual importance exists in 3D computer vision. With this objective, we present PIE-NET, a deep neural network which is trained to extract parametric curves from a point cloud that compactly describe its edge features. We demonstrate that a *region proposal* network architecture can already significantly outperform the state-of-the-art, which includes both traditional (non-learning) methods, and a recently developed deep model [10], the only other learning-based edge-point classifier, to the best of our knowledge.

Our contribution does not lie in merely surpassing the state-of-the-art from traditional graphics and geometry processing techniques. More importantly, we are introducing a *new learning problem* to the machine learning community, as well as defining the corresponding challenge for the recently published ABC dataset [4]. We also show that, yet again, networks can perform excellently even without resorting to excessive amounts of domain-specific (i.e., differential geometry) knowledge, as the only domain knowledge we assume is a simple curve parameterization.

## Acknowledgement

We thank the anonymous reviewers for their valuable comments. This work was supported in part by National Key Research and Development Program of China (2019YFF0302902 and 2018AAA0102200), National Natural Science Foundation of China (61572507, 61532003, 61622212, U1736217 and 61932003).

## Broader impact

We introduce a methodology that can benefit a range of current and new applications, from 3D data acquisition to object recognition. On the broader societal level, this work remains largely academic in nature, and does not pose foreseeable risks regarding defense, security, and other sensitive fields.

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
