[Reviews · NeurIPS 2020]

Review 1

Summary and Contributions: In this paper, the authors present PIE-Net, a neural network based approach to edge detection in point cloud. This end-to-end approach is a three module detector: a edge/corner detector at point level, a curve proposal network and curve proposal selector. Three type of curves (the most common in the dataset) are handled by the approach: segment, circles and B-splines. The method is evaluated on the ABC dataset for edge detection and outperforms the state of the art.

Strengths: - Edge/Corner detection and curve parametrization is a well known problem if surface reconstruction / shape abstraction. - The end-to-end detection and estimation process is an apealing idea. The three-step process is sounded. - PIE-Net is the first deep method that directly estimates the parametric curves. - PIE-NET outperforms the state of the art by a large margin on the ABC dataset for edge detection.

Weaknesses: - The paper is not self-contained. Lines 133/144: the paper shuld be self contained and no explicit reference to the supplementary material shall be find in the text. The author have used the supplementary material for section 6 and 7 which describes essential components of the method: loss function, curve proposals, curve selection. - My main concern is the relevance for the NeuRIPS community which is usually more focused on machine learning than computer graphics / computational geometry. The interest for the geometry commmunity is strong. However the machine learning side seems not to be the objective of the paper: close curve loss is described in the supplementary material. The parameter $\tau_c$ and $\tau_e$ are not really discussed.

Correctness: .

Clarity: The paper is well written and easy to understand (except for missing parts added to the supplementary material). Figures are relevant and illustrate well the text.

Relation to Prior Work: The edge estimation in point cloud with deep learning is a fairly new topic. The related work concerning other feature estimation like normals or curvature could be more developped but that is not critical.

Reproducibility: Yes

Additional Feedback: ### Typos - line 154: L should be a cursive font ### Other comments - Ablation study: Sphere radius: wouldn't it someting like $R'=(1+\epsilon)R$ be good radius for the circle? when noise points of the edge could fall outside the sphere? - Ablation study: $\tau_c$ and $\tau_e$: In classification one would expect the threshold be 0.5, not hand crafted. What is the cause of this "overlearning"? Should the NMS be sufficient to compensate the innacuracies? - Number of sampled points: 8096 points. How does it perform with less points / random number of points / non uniform sampling ? - The use of the nearest neighbor assigment for ground truth generation make it dependent on the number of sampled points: less point generates larger edges. How would you circumvent this ? ### Recommendation Despite the strong benchmark results, I rate the paper to "marginally below acceptance threshold" (cf weaknesses). To me the paper should include part of (if not all) the supplementary material. Moreover, even though the task / approach to the task is new, the novelty on the machine learning side is limited. ### Update after rebuttal Response are convincing, the quality of the paper is good. I rate the paper to 6.


Review 2

Summary and Contributions: The paper proposes a new interesting method to detect edges in point clouds. The edges are output as parametric curves (lines/b-splines/circles). The method works in three steps: 1. point classification (edge x corner x none) 2. curve proposal generation 3. proposal selection The method heavily builds upon PointNet++, which is used for multiple purposes.

Strengths: The proposed method is new, its exposition sound, and would be relevant for the NeurIPS community. The method clearly outperforms alternative methods, including EC-Net. Also the provided evaluation of the method is thorough enough (e.g., using a large CAD models dataset) and well executed (e.g., random edge-detection results shown) showing promising results of the method.

Weaknesses: The only important thing I am really missing in the paper is an evaluation of the runtime of the method. Especially given that all pairs of detected corners are considered for the generation of open curve proposals. A clear limitation, pointed out by the authors, is that the presented version of the method uses a limited set of geometric primitives, e.g., ellipses are not included. However, the reasons for this are explained and reasonable. It would be useful if authors could add a sentence or two about how this could be patched and if they would expect some difficulties. Another downside is the method involves relatively many ad-hoc/heuristic bits. Specific points: - I find it unfortunate that important parts of the paper are in the supplementary. The closed curve proposal generation is at least referred to from the main text, however, the proposal selection is just omitted without mention. There should be at least a sentence pointing to the supplement. - An explanation of the clustering (L129) could be improved, as of now I would reimplement it using Euclidean distance and a maximum linkage threshold of \delta. Please check this is the intended way and explain better otherwise.

Correctness: The claims in the paper as well as the methodology used for evaluating the method seem sound to me. In Sec. 4.3, please check the noise perturbation, I would guess the range should be [1 - X, 1 + X], not [-1 - X, 1 + X].

Clarity: The writing is clear and the method is well explained. It was a pleasure to read. Minor comments: - L174: corners -> corner - Section 4.1 "Ablation studies" is more of an "Selection of parameter values", the current heading is slightly misleading - L250: learning-base -> learning-based

Relation to Prior Work: Yes, the differences to the state-of-the-art method, EC-Net, are discussed. The proposed method is also benchmarked against EC-Net.

Reproducibility: Yes

Additional Feedback: I would like to thank the authors for the additional information and clarification.


Review 3

Summary and Contributions: The authors propose an end-to-end learnable technique to robustly identify feature edges in 3D point cloud data. The edges are represented as a collection of parametric curves. Experimental results show that the proposed method is significantly better than the state-of-the-art and generalizes well to novel shape categories.

Strengths: 1. The idea of treating parametric edge inference as a region proposal task is interesting. 2. The model uses a deep network to detect edges, corners and performs curve proposal generation.

Weaknesses: 1. The novelty of the paper is limited. To my knowledge, this paper is just a combination of several existing approaches, such as PointNet, non-maximal suppression, etc. 2. The application of this method is also limited. The authors only evaluate on one dataset, which limits the application of the PIE-NET.

Correctness: Yes

Clarity: Yes. The writing is clear.

Relation to Prior Work: Yes

Reproducibility: Yes

Additional Feedback:


Review 4

Summary and Contributions: This paper introduces the novel (for deep learning) problem of edge and corner detection on 3D point clouds, which has clear motivation in terms of future application and relation to 2D vision. The paper approaches this problem by first classifying whether each point belongs to a corner or edge. Each pair of corners is then considered to propose a edge, which is given by a segmentation mask of which points belong to it, what type of edge it is (e.g. line or curve), and its closed-form parameterization. This framework surpasses previous state-of-the-art non-learning based methods.

Strengths: The novelty of the task and its potential usage in downstream tasks such as scene understanding are apparent. The method is simple and straightforward, which I consider a strength given that the goal of the paper is to essentially build a edge-detection module that can be plugged into a variety of downstream tasks. The experiments are convincing in showing that a learned edge detector is superior to previous non-learning based geometric methods. Moreover, generalization to new object categories, which is commonly a disadvantage of learning-based methods compared to non-learning methods, is not a big issue for this method. Code and data are available and easy to read.

Weaknesses: Based on the code, it seems that the point segmentation and edge proposal module are trained separately. Please make this clear in the main text. As pointed out by other reviewers, the central contribution of this paper would be of use towards the graphics/vision commuity rather than the machine learning community. However, I still believe that the paper is worthy of acceptance at Neurips.

Correctness: Experiments are thorough and convincing.

Clarity: Yes

Relation to Prior Work: Yes

Reproducibility: Yes

Additional Feedback: Is there a reason an end-to-end method wasn't used (i.e. one stage of training)? After reading the rebuttal and discusion with reviewers, I maintain that the topic of this paper is suitable for publication at Neurips and the quality is well above the publication threshold.

[Author Response · NeurIPS 2020]

We thank all the reviewers for their insightful comments and encouraging remarks. Our rebuttal addresses the main questions, but not all, due to space limitations; the minor questions will all be dealt with in the revision.

**ML contribution (R1 & R3).** Our paper is positioned as an *application* paper, which constitutes a novel application of machine learning techniques to one of the most classic problems in 3D shape perception. Specifically, "PIE-Net is the first deep method that directly estimates the parametric curves (R1)", and produces very promising results.

Our innovation is not intended to be the development of new ML techniques, but lends itself to the application setting. Also, as a first attempt, we would rather showcase a relatively simple approach, rather than a complex one. Our work shows that even a simple learning paradigm such as a region proposal network can already significantly outperform the state of the art. Hence, the value of our contribution is not as a "final say", but in setting up a strong baseline to entice and stimulate future work on a fundamental and frequently encountered task in shape understanding.

**"Only evaluated on one dataset, limiting applications (R3)."** Sorry, this is not quite true. Please refer to Fig. 8 in the paper (and more results in the supplementary), where we applied PIE-Net trained on ABC to test shapes belonging to *novel* categories, i.e., categories *not found* in the training set. These results are exactly meant to demonstrate the generality and superiority of PIE-Net, even on non-CAD models such as the vases in the last two rows of Fig. 8. In terms of *quantitative* results, R3 is correct in that we only tested on ABC as the ground truth is available.

**Paper should be more self-contained (R1 & R2).** This is quite easy to fix. Clearly, the issue was limited space, as we wanted to show more experimental and comparison results in the paper. In the revision, we can make some space and move condensed coverage of necessary technical details from the supplementary material to the main paper.

**"The only important thing I am really missing is an evaluation of the runtime of the method (R2)."** For all the results shown in the paper, the average running time is about 0.5 second for point classification and 3 seconds for curve generation, per point cloud. In comparison, the average running times for point classification by VCM, EAR, and EC-Net are 5.5, 4.0, and 0.8 seconds, respectively — they are all slower than PIE-Net.

Training times for the point classification, the open and closed curve proposal networks were about 23, 12, and 8 hours, respectively, for 100 epochs, on an NVIDIA TITIAN X GPU. We will add all these numbers to the revision.

**"$\tau_c$, $\tau_e$: In classification one would expect the threshold be 0.5, not hand crafted (R1)."** These thresholds are not learned (hence no "overlearning"); they are set by the user. NMF was employed to select among points that all passed the threshold. While 0.5 is a typical classification threshold, it is far from a "fit-all" choice; it is often unsuitable in the case of *imbalanced classification*, which is our case here. In general, choices of thresholds are problem-dependent.

**Performance on random/non-uniform sampling (R1).** We sampled 100K points uniformly over each CAD shape, and then sub-sampled 8,096 points non-uniformly via random sampling. We re-trained and re-tested PIE-Net on the new point clouds, keeping all other settings unchanged. The following shows the performance numbers on metrics listed in Fig. 7 in the submission; the original numbers from Fig. 7 are provided in brackets for reference:

ECD: 0.0137 (0.0088); IOU: 0.5976 (0.6223); precision: 0.6816 (0.6918); recall: 0.8319 (0.8584).

As we can see, while there is a slight performance degradation, the new numbers are quite comparable to the original and still outperform all performance statistics obtained by VCM, EAR, and EC-Net, except for one case, VCM with $\tau = 0.12$, which yielded a recall of 0.8385, but it is paired with a very low precision of only 0.3063.

**"Point segmentation and edge proposal modules are trained separately ... why not end-to-end (R4)?"** We assume that R4 meant that our point *classification* and curve proposal networks were trained separately, which is correct. We will make sure to clearly state that in the revision as requested by the reviewer.

While end-to-end learning holds many merits, there is also a "flip side", e.g., see recent discussions on the limits to end-to-end learning [1, 2]. In our case, while it could be possible to design an end-to-end trained network to perform PIE-Net's tasks, we believe that it will likely be an inefficient approach. The network may be overly complex with a higher-than-necessary capacity, hence prone to overfitting. Our view is that point classification and parametric curve generation are standalone and clearly delineated modules, where attaining the best results for each task individually does not hinder the final outcome when the two modules are executed sequentially. Perhaps another perspective of our design decision is that we are utilizing the inductive bias arising from the pursuit of model simplicity.

# References

[1] Tobias Glasmachers. Limits of end-to-end learning. *Proceedings of Machine Learning Research*, 77:17–32, 2017.

[2] Francesco Locatello, Stefan Bauer, Mario Lucic, Gunnar Rätsch, Sylvain Gelly, Bernhard Schölkopf, and Olivier Bachem. Challenging common assumptions in the unsupervised learning of disentangled representations. In *ICML*, 2019.


[Meta-Review · NeurIPS 2020]

The reviewers felt that this paper provides an interesting and novel approach. The demonstrated approach outperforms the previous state of the art by a large margin. The main concern by reviewers is whether this paper is appropriate for a machine learning conference. All reviewers agree that the submission is a strong application paper that would be a strong submission for a computer vision, graphics, or computational geometry conference. However, the reviewers questioned whether the machine learning novelty in this paper is sufficient for a machine learning conference. At the same time, the reviewers noted that the NeurIPS CFP (Call for Papers) explicitly lists Computer Vision Applications of ML and they noted that the proposed submission has novelty as an application of ML. The reviewers also suggested that some items in the supplement be moved to the main text.